# The Role of Chloroplast Gene Expression in Plant Responses to Environmental Stress

**DOI:** 10.3390/ijms21176082

**Published:** 2020-08-24

**Authors:** Yi Zhang, Aihong Zhang, Xiuming Li, Congming Lu

**Affiliations:** 1State Key Laboratory of Crop Biology, College of Life Sciences, Shandong Agricultural University, Taian 271018, China; zhangyi@sdau.edu.cn (Y.Z.); ahzhang@sdau.edu.cn (A.Z.); 2State Key Laboratory of Crop Biology, College of Horticulture Science and Engineering, Shandong Agricultural University, Taian 271018, China; lixiuming@sdau.edu.cn

**Keywords:** chloroplast gene expression, photosynthesis, environmental stress response, transcription, RNA metabolism, translation

## Abstract

Chloroplasts are plant organelles that carry out photosynthesis, produce various metabolites, and sense changes in the external environment. Given their endosymbiotic origin, chloroplasts have retained independent genomes and gene-expression machinery. Most genes from the prokaryotic ancestors of chloroplasts were transferred into the nucleus over the course of evolution. However, the importance of chloroplast gene expression in environmental stress responses have recently become more apparent. Here, we discuss the emerging roles of the distinct chloroplast gene expression processes in plant responses to environmental stresses. For example, the transcription and translation of *psbA* play an important role in high-light stress responses. A better understanding of the connection between chloroplast gene expression and environmental stress responses is crucial for breeding stress-tolerant crops better able to cope with the rapidly changing environment.

## 1. Introduction

Plant often face environmental conditions that are unfavorable for growth and development. These adverse environmental conditions include abiotic and biotic stresses, such as drought, heat, cold, salt, and pathogen infection [1,2,3,4,5,6]. Environmental stresses pose a great threat to agriculture by limiting crop yields and productivity. The adverse effects of environmental stresses are getting worse due to the increasing worldwide population and climate change. To deal with these environmental stresses, plants rely on their ability to sense and cope with these stresses by regulating the expression of stress-responsive genes in the nucleus, cytoplasm, and organelles.

The chloroplast, a unique plant organelle, is the site of photosynthesis, intracellular signaling, and the production of various compounds important in metabolism, such as amino acids, hormones, nucleotides, vitamins, lipids, and secondary metabolites [7,8,9]. Chloroplasts also serve as sensors of the external environment. Under stress conditions, chloroplasts send messages to the nucleus through plastid-to-nucleus retrograde signaling, thus optimizing nuclear gene expression based on physiological requirements [7,8]. To date, several possible retrograde signaling pathways have been proposed, including pathways involving intermediates in tetrapyrrole biogenesis [10], the redox state of plastids [11,12], reactive oxygen species [13,14], secondary metabolites in chloroplasts [15,16], and chloroplast gene expression [17,18,19].

Chloroplasts are semi-autonomous organelles that have retained their own genomes. However, during evolution, most chloroplast genes were lost or transferred to the nucleus: On average, the chloroplast genomes of land plants have retained only 120 genes [20,21]. Nonetheless, these relatively few genes play fundamental roles in chloroplast activities such as energy production and gene expression [22]. Gene expression in chloroplasts is a highly complicated process, far more complex than in their prokaryotic ancestors. This is because chloroplasts have retained a hybrid gene-expression system that combines features of the prokaryotic gene-expression apparatus with eukaryotic innovations (e.g., RNA editing and RNA splicing), and its nascent polycistronic transcripts must undergo many post-transcriptional processing steps [22,23,24,25].

The proper expression of chloroplast genes is crucial for chloroplast development and photosynthesis. During the past decade, much effort has been invested in exploring the molecular mechanisms regulating chloroplast gene expression using genetic approaches. Many nucleus-encoded proteins involved in regulating chloroplast gene expression have been identified. However, studies of mutants of these proteins have shown that these mutants are also sensitive to various environmental stresses [8,26,27,28,29]. These findings suggest that there is a link between chloroplast gene expression and environmental stress responses, but less attention has been paid to this issue. In this review, we discuss the emerging roles of chloroplast gene expression in plant responses to environmental stresses.

## 2. The Characteristics of Chloroplast Gene Expression

The chloroplast gene-expression system is evolutionarily derived from photosynthetic bacteria that were endocytosed by ancestral eukaryotic plant cells more than 1.5 billion years ago [30]. During evolution, chloroplasts have retained core components of the gene-expression apparatus from their prokaryotic progenitors. In addition, they obtained many eukaryotic properties, such as RNA editing, the prevalence of introns, and complex processing patterns from polycistronic RNA precursors [31]. Here, we briefly describe the processes of chloroplast gene expression in plants (Figure 1).

### 2.1. Transcription

In plants, chloroplast gene transcription is conducted by two distinct types of RNA polymerases: Nucleus-encoded RNA polymerase (NEP) and plastid-encoded RNA polymerase (PEP) [32,33]. In mature chloroplasts, PEP represents the major transcriptional machinery, which transcribes >80% of all primary chloroplast transcripts, while NEP transcribes chloroplast housekeeping genes [34]. NEP is a phage-type RNA polymerase with a single subunit. In Arabidopsis (*Arabidopsis thaliana*), NEP is encoded by two nuclear genes, *rpoTp* and *rpoTmp* [35]. PEP is a bacteria-type RNA polymerase composed of four core enzyme subunits (α, β, β′, and β″) and a promoter-recognizing subunit (σ factor). The core enzyme subunits of PEP are encoded by a set of genes located in the plastid genome: *rpoA*, *rpoB*, *rpoC1*, and *rpoC2* [33]. By contrast, during evolution, genes for σ factors, which provide the necessary promoter specificity to PEP, were transferred to the nuclear genome, perhaps allowing the nucleus to regulate chloroplast gene transcription in response to environmental and developmental cues [36]. PEP and a set of polymerase-associated proteins (PAPs) form a huge protein complex required for transcription. All PAPs are encoded by genes in the nucleus, and most of them are the components of plastid transcriptionally active chromosome (pTAC) [37]. These PAPs are predicted to be involved in DNA and RNA metabolism (PAP1/pTAC3, PAP2/pTAC2, PAP3/pTAC10, PAP5/pTAC12, PAP7/pTAC14, and PAP12/pTAC7), redox regulation from photosynthesis (PAP6/FLN1, PAP10/TrxZ, and PAP12/pTAC7), and protecting the PEP complex from reactive oxygen species (PAP4/FSD3 and PAP9/FSD2) [38]. The transcriptional regulation of chloroplast genes is essential for the proper functioning of chloroplasts and for overall plant growth under both normal and adverse conditions.

### 2.2. RNA Metabolism

Most chloroplast genes in plants are organized as operons. These polycistronic primary RNAs require extensive processing, including 5′ and 3′ trimming, intercistronic cleavage, RNA splicing, and RNA editing [39]. Evidence suggests that 5′ and 3′ trimming and intercistronic cleavage are important for moderating RNA stability and translation within chloroplasts [40,41,42,43,44]. In plants, approximately 20 chloroplast genes (encoding proteins or structural RNAs) are interrupted by introns. RNA splicing removes the intron sequences of genes from primary transcripts to enable the production of mature mRNA with the correct genetic information [45,46]. RNA analyses have shown that RNA editing (mainly in the form of C-to-U base conversions) is highly widespread within the chloroplasts of land plants. During this process, numerous C-to-U conversions alter the coding sequences of chloroplast mRNAs, regulate RNA secondary structures that influences the splicing and/or stability of RNAs, or generate translational start sites (AUG) [47,48]. All of these RNA metabolic events depend on many nucleus-encoded proteins, most of which likely arose during coevolution between the host and endosymbiont. For detailed information on chloroplast RNA metabolism, we direct the reader to recent reviews in this area [39,43,45,48,49].

### 2.3. Translation

Chloroplasts possess a bacterial-type 70S ribosome as well as a full set of transfer RNAs (tRNAs) and ribosomal RNAs (rRNAs), which conduct protein translation [50]. The 70S ribosome comprises two multi-component subunits: The large (50S) and small (30S) subunits. Both subunits contain rRNAs and various plastid- and nucleus-encoded proteins [51,52,53,54,55].

In general, the chloroplast ribosome has a bacterial-type structure, but with some distinctive features. Chloroplast ribosomes contain the complete set of bacterial-type rRNAs (23S, 16S, and 5S rRNA) with functions analogous to those in bacteria. For example, 23S rRNA exhibits peptidyl transferase activity, whereas 16S rRNA functions as the decoding center and serves as a scaffold for other proteins during ribosome assembly [56,57]. However, the chloroplast contains an additional 4.5S rRNA not found in bacteria that is homologous to the 3′ end of prokaryotic 23S rRNA, suggesting that it was derived from fragmentation of this prokaryotic rRNA [58]. Additionally, two post-transcriptional cleavage sites within the 23S rRNA precursor generate mature 23S rRNA fragments. All of these fragments are assembled into the mature 70S ribosome and combined via intermolecular base pairing [52]. During evolution, obvious changes also occurred in the protein composition of the chloroplast ribosome. The homologs of bacterial proteins Rpl25 and Rpl30 were completely lost in chloroplasts [59]. Several new components of the plastid (chloroplast) ribosome, known as plastid-specific ribosomal proteins (PSRPs), have also been identified [59,60]. PSRP5 and PSRP6 in the 50S subunit and PSRP2 and PSRP3 in the 30S subunit are believed to be intrinsic components of the chloroplast ribosome [52].

## 3. Chloroplast Gene Expression and Environmental Stress

To date, genetic analyses have revealed many nucleus-encoded proteins that regulate not only chloroplast gene expression but also responses to environmental stresses. Functional analyses of these nucleus-encoded proteins have indicated that chloroplast gene expression is involved in plant responses to environmental stresses (Table 1).

### 3.1. Transcription and Environmental Stress Responses

The transcriptional regulation of chloroplast gene expression is crucial not only for photosynthesis but also for plant development. Recent studies have revealed that the transcriptional control of chloroplast gene expression also plays important roles in plant responses to environmental changes. The chloroplast gene *psbA* encodes the D1 reaction center protein of photosystem II (PSII) [61,62,63]. Due to the nature of PSII photochemistry, D1 protein is continuously subjected to photodamage, which decreases photosynthetic activity (an effect known as photoinhibition). These damaged D1 proteins are replaced by *de novo* synthesized D1 proteins following the partial disassembly of the PSII complex [64,65]. Hence, the capacity to repair photodamaged PSII strongly depends on the ability to generate new D1 protein. Chloroplasts can adjust the transcriptional efficiency of *psbA* during photoinhibition under adverse environmental conditions such as high light and temperature [66,67,68]. During chloroplast evolution, several nucleus-encoded proteins have developed the ability to regulate *psbA* transcription in order to repair photodamaged PSII under adverse environmental conditions. Tomato (*Solanum lycopersicum*) WHIRLY1 (SlWHY1) was recently found to upregulate *psbA* transcription under chilling conditions. Under these conditions, the chloroplast-localized SlWHY1 promotes the transcription of *psbA* by directly binding to the upstream region of its promoter (the sequence “GTTACCCT”), resulting in increased D1 abundance to relieve photoinhibition [69,70]. Overexpression of *SlWHY1* leads to increased *de novo* synthesis of D1 protein and increased resistance to photoinhibition under chilling conditions [69]. These findings suggest that *psbA* transcription is an important target for regulating PSII activity to adjust plant resistance to environmental stresses.

The chloroplast gene *PsbD* encodes the reaction center protein D2 of PSII [86]. The expression of *psbD* is controlled by four PEP promoters. One of these is the blue-light-responsive promoter *psbD* BLRP [87,88,89]. The structure of *psbD* BLRP is distinct from that of common PEP promoters, which are characterized by conserved −35 and −10 elements. The *psbD* BLRP contains three *cis*-elements, including the AAG box, PGT box, and −10 element, but lacks the conserved −35 element [32,89,90]. This promoter has been well characterized. *psbD* BLRP transcription is specifically regulated by chloroplast-localized sigma factor 5 (SIG5) [71,91]. *psbD* BLRP transcription is also induced by environmental stresses, such as high salinity, low temperature, and osmotic stress [71]. In addition, *psbD* BLRP transcription is modulated in response to the relative proportions of red and far red light in a process mediated by signals from phytochromes [92]. Thus, *psbD* BLRP transcription is modulated during plant responses to environmental stress and sensing of light signals. Indeed, high *psbD* BLRP activity favors the synthesis of D2, thus relieving high-light-induced damage to PSII [93]. On the other hand, *psbD* BLRP transcription mediated by SIG5 shows obvious circadian oscillation, revealing how chloroplast gene expression is involved in the circadian oscillator [94].

*psbD* BLRP transcription may be also involved in biotic stress responses. Pathogens deliver various effectors into plant host cells when pathogens attack plants. These effectors assist pathogen proliferation and suppress plant defense responses [95,96,97,98,99,100,101]. Two *Pseudomonas* effectors, HopR1 and HopBB1, has been suggested to be involved in *psbD* transcription by targeting PTF1 (PLASTID TRANSCRIPTION FACTOR 1), a transcription factor for *psbD* BLRP transcription [102,103]. Moreover, the loss of PTF1 leads to more resistant to *Pseudomonas syringae* pv. *tomato* strain DC3000 in Arabidopsis [104]. Thus, *psbD* BLRP transcription may play a role in biotic stress responses.

The *psbEFLJ* operon contains four chloroplast genes: *psbE*, *psbF*, *psbL*, and *psbJ*. These genes encode the α and β subunits of cytochrome b559, PsbL, and PsbJ, respectively, which are crucial for the proper functioning of PSII [105,106]. The transcriptional regulation of *psbEFLJ* was recently investigated. *psbEFLJ* transcription is positively regulated by the nucleus-encoded protein mTERF5 (mitochondrial Transcription Termination Factor 5), which acts as a pausing factor [72,107]. mTERF5 causes transcriptional pausing on *psbEFLJ* by binding to the nucleotides +30 to +51 from the transcription start site and recruits additional pTAC6 into the PEP complex at the pausing region to form an enhanced PEP complex, thus positively regulating *psbEFLJ* transcription. In addition, *mterf5* mutants are less sensitive to NaCl and abscisic acid (ABA) than wild-type plants, indicating that mTERF5 functions as a negative regulator of salt tolerance, perhaps via ABA signaling [73]. These findings point to functional links between *psbEFLJ* transcription and salt tolerance as well as ABA signaling.

### 3.2. RNA Metabolism and Environmental Stress Responses

RNA metabolism in chloroplasts is remarkably complex, involving a series of steps such as 5′ and 3′ trimming, RNA editing, splicing, and intergenic cleavage [31]. Analyses of mutants with defective RNA editing suggested that RNA editing, splicing, and stability help regulate environmental stress responses in plants [28,29].

An overall deficiency in chloroplast RNA editing (C-to-U base conversion) in Arabidopsis could be caused by the mutation of *ORRM1* (*Organelle RRM Protein 1*), encoding an essential plastid RNA editing factor. *orrm1* mutants exhibited greatly reduced RNA editing efficiency compared to wild-type Arabidopsis at 62% (21 of 34) of the chloroplast editing sites. Among these, the editing efficiency at 12 sites decreased by at least 90%, whereas that of the nine other sites decreased by 10% to 90% in *orrm1* vs. wild-type plants [74]. The reduced RNA editing deficiency at multiple sites in *orrm1* plants did not result in distinctive phenotypes at normal temperatures (22 °C), but the mutants were sensitive to chilling, displaying yellow emerging leaves under chilling conditions (4 °C) [75]. These findings suggest that chloroplast RNA editing confers low-temperature tolerance in Arabidopsis. However, the RNA editing site that confers this improved low-temperature tolerance is unknown.

The *indica* (*Oryza sativa* ssp. *indica*) rice cultivar *Dular*, referred to as *dua1*, is planted in tropical regions of Southeast Asia, including India and the Philippines. *dua1* plants are less tolerant of low temperatures than Nipponbare (*O. sativa* ssp. *japonica*) plants, which are grown in northern areas of Asia, as *dua1* plants display pale leaves under low-temperature conditions (19 °C). A recent study revealed that that the low-temperature sensitivity of *dua1* is caused by defective RNA editing of the plastid ribosome gene *rps8*, which is located 182 nt downstream of the translational start site (*rps8*-182). The edited *rps8* transcripts generate RPS8 protein with altered amino acid hydrophobicity, suggesting that RNA editing at *rps8*-182 improves low-temperature tolerance in rice by moderating the stability of RPS8 protein under low-temperature conditions [76]. Chloroplast genomes have very slow rates of sequence evolution, averaging ~5-fold slower than nuclear genomes [108,109], suggesting that chloroplast RNA editing evolved to improve low-temperature tolerance by increasing protein stability.

*ndhB* encodes the B subunit of the chloroplast NADH dehydrogenase-like complex that is required for cyclic electron flow around photosystem I [110,111]. The defective RNA editing of *ndhB*-2, *ndhB*-3, *ndhB*-4, and *ndhB*-6 sites enhances the disease resistance against fungal pathogens in Arabidopsis [112]. This finding suggests that chloroplast RNA editing is interlinked with plant immunity.

*rpl2* encodes a component of the 50S subunit in the chloroplast ribosome. This gene contains only a group II intron. In rice, the splicing of this intron is specifically regulated by WHITE STRIPE LEAF (WSL), a pentatricopeptide repeat (PPR) protein. Compared to the wild type, *wsl* mutants exhibit a decreased germination rate and reduced shoot and root growth upon treatment with ABA but not with α-naphthaleneacetic acid (NAA, an auxin), gibberellic acid (GA), epi-brassinosteroid (BL), or 6-benzylaminopurine (6-BA, a cytokinin). This finding suggests that the ABA signaling process is specifically affected in *wsl*. These mutants also display decreased germination rates when grown on medium supplemented with sugar and NaCl [77]. Sugar and salinity responses are closely connected with ABA signaling, and several ABA-related genes (e.g., *ABI3* and *WRKY24*) are induced by ABA treatment in *wsl* mutants, suggesting that *rpl2* splicing plays an important role in plant responses to ABA.

In Arabidopsis, the splicing of chloroplast *trnA*, *trnI*, *rpl2*, *rps12* intron 1, and *rps12* intron 2 is regulated by DEAD-BOX RNA HELICASE 3 (RH3) [78]. Null mutants of *RH3* are embryo lethal, whereas the weak allele *rh3-4* displays retarded plant growth and pale-green leaves, along with considerable decreases in the splicing efficiency of *trnA*, *trnI*, *rpl2*, *rps12* intron 1, and *rps12* intron 2. Moreover, the endogenous ABA contents of 1-week-old *rh3-4* seedlings are ~50% lower than those of wild-type plants, suggesting that RH3 plays a role in ABA biosynthesis. The mutation of *RH3* results in the reduced expression of nucleus-encoded gene *ABA1* and *NCDE4*, encoding two crucial enzymes of the ABA biosynthetic pathway, perhaps explaining the decreased ABA contents of *rh3-4* seedlings. Consistent with their decreased ABA contents, *rh3-4* mutants exhibit more severely inhibited plant growth and greening than the wild type under abiotic stress conditions including salinity, cold, and dehydration stress [78,79,80]. These findings suggest that chloroplast RNA splicing of these genes is required for environmental stress responses in plants, especially responses related to ABA signaling. Yet how chloroplast RNA splicing regulates environmental stress responses is currently unknown. A defect in chloroplast RNA splicing would be likely to result in defective photosynthetic performance, thus leading to enhanced sensitivity to environmental stresses. Alternatively, chloroplast RNA splicing might trigger plastid-to-nucleus retrograde signaling to regulate plant stress responses.

Chloroplast RNA stability is also crucial for the proper expression of chloroplast genes. Increasing evidence indicates that chloroplast RNA stability is involved in plant responses to environmental stresses. Chloroplast ribonucleoproteins CP31A and CP29A are RNA chaperone proteins that associate with large sets of chloroplast transcripts [81,113]. Arabidopsis mutants with deletions of *CP31A* and *CP29A* do not have unusual phenotypes under normal conditions but show bleaching of newly emerging leaves at the bases of the youngest leaves under cold stress (8 °C). Kupsch et al. demonstrated that CP31A and CP29A are required for the accumulation of transcripts of many chloroplast genes under cold stress (8 °C), such as *psaA*, *psbD*, *psbF*, *psbB*, *petB*, *ndhF*, and *rbcL*. This cold-sensitive phenotype could be explained by a decreased stability of chloroplast transcripts in the *cp31a* and *cp29a* mutants [81,113]. DEAD-box RNA helicase 22 (RH22) is another chloroplast RNA chaperone. In cabbage (*Brassica rapa*), *RH22* expression was significantly upregulated by drought, heat, salt, and cold stress but markedly downregulated by UV stress. The overexpression of cabbage *RH22* enhanced the stability of chloroplast transcripts and improved growth and survival in Arabidopsis under drought and salt stress [114]. Moreover, Arabidopsis plants overexpressing cabbage *RH22* displayed better growth and more green leaves upon ABA treatment than the wild type, along with decreased expression of *ABI3*, *ABI4*, and *ABI5*, suggesting that chloroplast RNA stability plays a part in ABA signaling pathways [114]. Chloroplast RNA stability might have a positive role in plant responses to environmental stress by enhancing the translation of chloroplast genes.

### 3.3. Translation and Environmental Stress Responses

Translation is the final step in chloroplast gene expression. Chloroplast gene translation regulates protein accumulation to optimize photosynthetic performance and to attenuate photooxidative damage. Thus, the regulation of chloroplast gene translation represents a unique component of plant responses to internal and external stimuli.

Most plants growing in direct sunlight routinely encounter high-light stress; the resulting high photon flux exceeds the photosynthetic capacity, thereby damaging the chloroplast. To explore the regulation of chloroplast gene translation during the rapid adaptation of plants to high light, a systematic ribosome profiling study was performed to detect changes in chloroplast gene translation efficiency in tobacco seedlings following transfer from moderate light to high light. The ribosome occupancy on *psbA* transcripts (encoding PSII reaction center protein D1) increased in response to high-light treatment [115]. Given that D1 protein is the main site prone to photodamage by high light, the upregulated *psbA* translation should substantially facilitate the repair of PSII under high-light stress. However, the molecular mechanisms underlying the translational activation of *psbA* under these conditions remain to be further explored.

Studies on the functions of chloroplast ribosome proteins have revealed that maintaining sufficiently high chloroplast gene translation efficiency is important for proper chloroplast development at low temperature. Maize (*Zea mays*) mutants with a loss of ribosomal protein RPS17 were pale green when grown at moderate temperature (27 °C) but appeared albino under cool conditions (17 °C) [83]. Tobacco mutants with a loss of the ribosomal protein Rpl33 showed no visible phenotypes at any stage of development under standard conditions, with similar development, growth rates, and onset of flowering to wild-type plants. However, the *Rpl33* knockout mutants were sensitive to cold stress, although not to heat or to low or high light levels. When *Rpl33* knockout mutants were transferred to cold-stress conditions (4 °C), they exhibited strong photooxidative damage symptoms and recovered much more slowly from low-temperature stress than wild-type plants [85]. As with Rpl33, the loss of the ribosomal protein Rps15 in tobacco resulted in a growth phenotype almost identical growth to that of wild-type plants, although young plants grew slightly more slowly and the onset of flowering was slightly delayed. However, the *Rps15* knockout mutants were cold sensitive, with more severe pigment loss and worse photosynthetic performance than wild-type plants [84]. Together, these findings suggest that the maintenance of plastid translational capacity is important in enabling plant tolerance to chilling stress.

In Arabidopsis, the expression of the chloroplast ribosome protein gene *RPS1* was considerably induced by heat stress (2 h at 38 °C). *RPS1* knockdown mutants (*rps1*) displayed retarded growth and slightly pale-green leaves. When *rps1* seedlings were exposed to transient high-temperature conditions (3 h at 45 °C), they were much more heat sensitive than wild-type seedlings, as almost no mutants survived after a 7-d recovery, whereas more than 90% of wild-type seedlings did. However, there were no significant differences between *rps1* and wild-type plants under osmotic and salinity stress. These results suggest that decreased *RPS1* expression alters cellular heat stress responses by disrupting chloroplast gene translation rather than through general physiological defects. RPS1 is required to activate the expression of *HsfA2* (*HEAT STRESS TRANSCRIPTION FACTOR A-2*), a highly heat-shock-inducible gene encoding a transcription factor that is crucial for triggering cellular responses to heat stress. The constitutive expression of *HsfA2* was sufficient to rescue the heat-sensitive phenotype of *rps1* mutants, suggesting that the defective expression of *HsfA2* is responsible for the heat-sensitive phenotype of *rps1* mutants. Like the *rps1* mutant phenotype, treatment with lincomycin, an inhibitor of chloroplast gene translation, also led to an obvious reduction in the expression of *HsfA2* in response to heat stress [82]. These findings reveal a plastid-to-nucleus retrograde signaling pathway that regulates chloroplast gene translational capacity to transcriptionally activate cellular heat stress responses, especially the HsfA2-dependent heat tolerance pathway.

## 4. Conclusions and Future Perspectives

Photosynthesis, one of the most important physiological processes in plants, is highly sensitive to environmental stresses. These stresses often inhibit photosynthesis considerably [116,117,118,119,120,121,122,123,124]. Many studies have reported that chloroplasts can act as sensors of the external environment [7,19]. Thus, in addition to hosting photosynthesis, chloroplasts play important roles in plant responses to various environmental stresses. Likewise, the proper expression of chloroplast genes is crucial for chloroplast development, photosynthesis, and plant development. However, as summarized in this review, many studies indicate that chloroplast gene expression is also important for plant stress responses.

Further elucidating the roles of chloroplast gene expression in plant responses to various environmental stresses would lay the foundation for genetically improving plant tolerance to the environment. However, the underlying molecular mechanisms remains largely unknown. Most studies have focused on the roles of nucleus-encoded proteins in regulating chloroplast gene expression and plant responses to environmental stress, while the direct connection between chloroplast gene expression and environmental stress responses has been largely ignored. Future studies should therefore address two major issues: How environmental stress triggers chloroplast gene expression, and what roles chloroplast gene expression plays in plant responses to environmental stress. Several specific issues need to be investigated. For example, D1 protein synthesis is important for PSII repair under high-light conditions [71,115,125,126,127,128]. Thus, enhanced *psbA* transcription and translation would help increase the tolerance of PSII to high light; however, it is still unclear to what extent high light triggers the transcription and translation of *psbA*. In addition, RNA editing, particularly C-to-U base conversion, is a widespread phenomenon in chloroplasts across nearly all plant species [48]. As discussed above, it is clear that RNA editing is important for plant tolerance of low temperature [74,75,76], yet the exact editing sites, and how RNA editing at these sites improves plant tolerance to low temperatures, remain to be explored. Finally, translation and splicing of several chloroplast genes trigger plastid-to-nucleus retrograde signaling and ABA signaling [73,77,79,80,114], but it remains unclear how the signals generated from these processes are transferred out of chloroplasts and integrated into these signaling pathways.

New technology is needed to better investigate the molecular mechanism of chloroplast gene expression in response to the environment. Chloroplast transformation is an extremely time-consuming and difficult process that has only been achieved in a few plant species, such as lettuce, poplar, and Arabidopsis. This make it extremely difficult to control chloroplast gene transcription using traditional genetic engineering approaches. To further explore the connection between chloroplast gene transcription and environmental stress responses, novel tools must be designed that are similar to CRISPR/Cas9 and RNA interference to knock out and knock down chloroplast genes. It is also critical to design an artificial RNA editing system to carry out RNA editing of specific sites in chloroplasts. This system would be useful for investigating the specific roles of different RNA editing sites in plant responses to environmental stresses and even creating new RNA editing sites that could improve stress tolerance in plants.

## Figures and Tables

**Figure 1 ijms-21-06082-f001:**
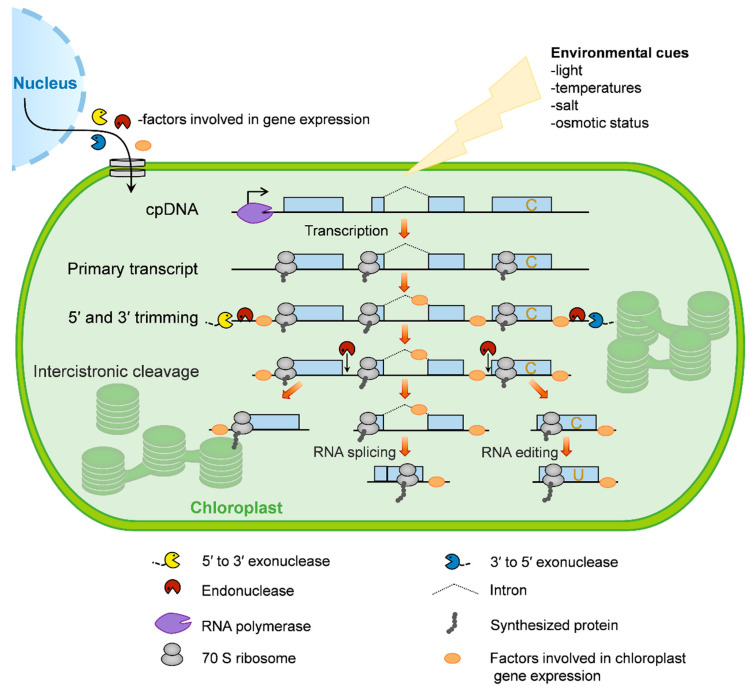
Overview of chloroplast gene expression. In plants, most chloroplast genes are organized as operons and are controlled by single promoters (bent arrow). These genes are transcribed by two distinct types of RNA polymerase: Nucleus-encoded RNA polymerase (NEP) and plastid-encoded RNA polymerase (PEP). The resulting primary transcripts require several processing steps to form mature mRNA, including 5′ and 3′ trimming, intercistronic cleavage, RNA splicing, and RNA editing. In order for these events to take place, numerous nucleus-encoded proteins are translated in the cytosol and imported into the chloroplast, where they control and/or regulate chloroplast gene expression. Chloroplast gene translation is conducted by bacterial-type 70S ribosomes, which occurs cotranscriptionally. Since the mRNA turnover rate within chloroplasts is slow, most ribosomes function in posttranscriptional steps. Moreover, chloroplast gene expression is involved in responses to environmental cues.

**Table 1 ijms-21-06082-t001:** Chloroplast gene expression and stress response mutants.

Gene Symbol Accession No.	Species	Mutant	Mutant Stress Phenotype	Molecular Function	Reference(s)
**Transcription:**					
*SlWHY1* (*Solyc05g007100*)	*Solanum lycopersicu*	*slwhy1*	Hypersensitivity to chilling	Promotes transcription of tomato *psbA* under chilling conditions	[69]
*SIG5* (*AT5G24120*)	*Arabidopsis thaliana*	*sig5*	Hypersensitivity to salt stress and high light	Specifically controls *psbD* transcription in response to circadian rhythms, environmental stresses, and light signals	[71]
*mTERF5* (*AT4G14650*)	*Arabidopsis thaliana*	*mterf5*	Decreased sensitivity to salt, ABA, and osmotic stress; altered sugar responses	Serves as a transcriptional pausing factor; specifically regulates the transcription of chloroplast *psbEFLJ*	[72,73]
**RNA metabolism:**					
*ORRM1* (*AT3G20930*)	*Arabidopsis thaliana*	*orrm1*	Hypersensitivity to low temperature	Controls RNA editing of 62% (21 of 34) of chloroplast transcripts	[74,75]
*DUA1* (*J043016D20*)	*Oryza sativa*	*dua1*	Hypersensitivity to low temperature	Required for RNA editing of the *rps8*-182 site	[76]
*WSL* (*Os01g0559500*)	*Oryza sativa*	*wsl*	Hypersensitivity to ABA, salinity, and sugar with increased H_2_O_2_ levels	Required for splicing of chloroplast *rpl2*	[77]
*RH3* (*AT5G26742*)	*Arabidopsis thaliana*	*rh3-4*	Hypersensitivity to salt stress and low temperature; reduced ABA content	Involved in splicing of chloroplast *trnA*, *trnI*, *rpl2*, *rps12* intron 1, and *rps12* intron 2	[78,79,80]
*CP29A* (*AT3G53460*)	*Arabidopsis thaliana*	*cp29a*	Hypersensitivity to low temperature	Required for maintaining the stability of various chloroplast transcripts	[81]
*CP31A* (*AT4G24770*)	*Arabidopsis thaliana*	*cp31a*	Hypersensitivity to low temperature	Required for maintaining the stability of various chloroplast transcripts	[81]
**Translation:**					
*RPS1* (*AT5G30510*)	*Arabidopsis thaliana*	*rps1*	Heat-sensitive phenotype; perturbed HSF-mediated heat stress response	Component of chloroplast ribosome small subunit; involved in activating cellular heat stress responses	[82]
*RPS17* (*GRMZM2G038013*)	*Zea mays*	*hcf60-m1*	Cold-induced bleaching and seedling-lethal phenotype	Component of chloroplast ribosome small subunit	[83]
*Rps15* (*GeneID:800489*)	*Nicotiana tabacum*	*Δrps15*	Hypersensitivity to low temperature	Component of chloroplast ribosome large subunit	[84]
*Rpl33* (*GeneID:800444*)	*Nicotiana tabacum*	*Δrpl33*	Hypersensitivity to low temperature	Component of chloroplast ribosome large subunit	[85]

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
