# Peer review of "The Role of Chloroplast Gene Expression in Plant Responses to Environmental Stress"

_ijms, 2020, doi:10.3390/ijms21176082_

Round 1
Reviewer 1 Report
A very interesting summary of the involvement of chloroplasts in gene expression. A wide range of literature sources was used to obtain information - 114 sources. The manuscript provides both general information and specific mechanisms involved in genes localized in chloroplasts. This may not have been the topic of this article, but readers would certainly also be interested in biotic stressors and chloroplast gene involvement. I assume that there are not many such studies and perhaps it would be possible to add them to this manuscript?
The English language was scientific and understandable, there were some errors in the text (such as duplication of words), but they did not affect the perception of the text and the essence.
Author Response
Responses to Reviewer #1 comments
A very interesting summary of the involvement of chloroplasts in gene expression. A wide range of literature sources was used to obtain information - 114 sources. The manuscript provides both general information and specific mechanisms involved in genes localized in chloroplasts. This may not have been the topic of this article, but readers would certainly also be interested in biotic stressors and chloroplast gene involvement. I assume that there are not many such studies and perhaps it would be possible to add them to this manuscript?
RESPONSE:
We appreciate the reviewer for the comment. Indeed, there are not many studies about chloroplast gene expression and biotic stressors. We have tried to cite several references that describe possible role of chloroplast gene expression in biotic stress responses (see line182-188 and line 228-232).
The English language was scientific and understandable, there were some errors in the text (such as duplication of words), but they did not affect the perception of the text and the essence.
RESPONSE:
Yes, we have corrected some duplication of words and some errors, e.g. line 267, 271 (duplication) and line 36, 96-97 (spelling errors).
Reviewer 2 Report
The review gives a comprehensive overview on involvement of processes from the categories: chloroplast gene transcription, RNA metabolism and translation in various stress related responses.
Detailed comments:
Abstract:
I would suggest including an example for the involvement of chloroplast gene expression in environmental stress response.
L35 - needs to be corrected: tough-trough
L95 - needs to be corrected: have showed – have shown
L97: please explain the meaning of “moderate the RNA structure”
L258: needs to be corrected: Kupsch et al. The authors….
L332: however, it
332 is unknown how high light triggers … Better: to what extent
Author Response
Responses to Reviewer #2 comments
The review gives a comprehensive overview on involvement of processes from the categories: chloroplast gene transcription, RNA metabolism and translation in various stress related responses.
Detailed comments:
Abstract:
I would suggest including an example for the involvement of chloroplast gene expression in environmental stress response.
RESPONSE:
Yes, we have added an example (see line 17-18).
L35 - needs to be corrected: tough-trough
RESPONSE:
Yes, we have corrected it (see line 37).
L95 - needs to be corrected: have showed – have shown
RESPONSE:
Yes, we have corrected it (see line 96-97).
L97: please explain the meaning of “moderate the RNA structure”
RESPONSE:
Yes, we have explained it by re-writing this sentence (see line 99-100).
L258: needs to be corrected: Kupsch et al. The authors….
RESPONSE:
Yes, we have corrected it (see line 267).
L332: however, it
332 is unknown how high light triggers … Better: to what extent
RESPONSE:
Yes, we have corrected it (see line 349-350).